Evaluating whole-genome sequencing quality metrics for enteric pathogen outbreaks

Wagner Darlene D. 1 2
Carleton Heather A. 3
Trees Eija 4
Katz Lee S. gzu2@cdc.gov 3 5
1 Division of Viral Diseases, Centers for Disease Control and Prevention , Atlanta , GA , United States of America
2 Eagle Medical Services, LLC , Atlanta , GA , United States of America
3 Enteric Diseases Laboratory Branch, Centers for Disease Control and Prevention , Atlanta , GA , United States of America
4 Association of Public Health Laboratories , Silver Spring , MD , United States of America
5 Center for Food Safety, University of Georgia , Griffin , GA , United States of America
Souza Valeria
Electronic publication date: 2021 Nov 25
Publication date: 2021
Volume: 9
Electronic Location ID: e12446
Received 2021 May 11; Accepted 2021 Oct 18
Copyright: ©2021 Wagner et al.
Copyright year: 2021
Copyright holder: Wagner et al.
License: This is an open access article, free of all copyright, made available under the Creative Commons Public Domain Dedication. This work may be freely reproduced, distributed, transmitted, modified, built upon, or otherwise used by anyone for any lawful purpose.
License URL: https://creativecommons.org/publicdomain/zero/1.0/

Keywords: SNP, Read cleaning, Read healing, Multiheal, Assembly

Funding: The authors received no funding for this work.

==============================
Background

Whole genome sequencing (WGS) has gained increasing importance in responses to enteric bacterial outbreaks. Common analysis procedures for WGS, single nucleotide polymorphisms (SNPs) and genome assembly, are highly dependent upon WGS data quality.

Methods

Raw, unprocessed WGS reads from Escherichia coli, Salmonella enterica, and Shigella sonnei outbreak clusters were characterized for four quality metrics: PHRED score, read length, library insert size, and ambiguous nucleotide composition. PHRED scores were strongly correlated with improved SNPs analysis results in E. coli and S. enterica clusters.

Results

Assembly quality showed only moderate correlations with PHRED scores and library insert size, and then only for Salmonella. To improve SNP analyses and assemblies, we compared seven read-healing pipelines to improve these four quality metrics and to see how well they improved SNP analysis and genome assembly. The most effective read healing pipelines for SNPs analysis incorporated quality-based trimming, fixed-width trimming, or both. The Lyve-SET SNPs pipeline showed a more marked improvement than the CFSAN SNP Pipeline, but the latter performed better on raw, unhealed reads. For genome assembly, SPAdes enabled significant improvements in healed E. coli reads only, while Skesa yielded no significant improvements on healed reads.

Conclusions

PHRED scores will continue to be a crucial quality metric albeit not of equal impact across all types of analyses for all enteric bacteria. While trimming-based read healing performed well for SNPs analyses, different read healing approaches are likely needed for genome assembly or other, emerging WGS analysis methodologies.

Introduction

Whole genome sequencing (WGS) has become instrumental to studies of enteric bacteria ranging from outbreak detection to microbial ecology. WGS data are analyzed by single nucleotide polymorphisms (SNPs), genome assembly, and multi-locus sequence typing (MLST). SNPs enable high-resolution quantification of nucleotide variation among microbial isolates, often by mapping the reads of WGS data to a reference genome. SNP analyses have proven indispensable to solving several high profile foodborne, waterborne, and domesticated animal outbreaks (Crowe et al., 2017; Katz et al., 2013; Marshall et al., 2018).

Genome assembly brings overlapping WGS reads together into contiguous blocks representing the original genome sequence (i.e., contigs). Genome assembly serves as a first step for many pipelines which characterize genes involved in antibiotic resistance and virulence (Singh et al., 2018; Wang et al., 2017) as well as elucidating phylogenetic relatedness through average nucleotide identity (Cheleuitte-Nieves et al., 2018; Goris et al., 2007). Given that SNPs and genome assembly are widespread analysis procedures, they can strongly impact the reliability of WGS data interpretation for clinical microbiology and epidemiologic investigations. In turn, the quality of SNPs and assemblies depend upon the quality of the original WGS reads. WGS quality control is thus an ongoing challenge for the multiple stakeholders involved in outbreak responses and evolutionary biology of enteric bacteria (Moran-Gilad et al., 2015; Timme et al., 2018). Characterization of WGS quality artifacts and quality metrics will build additional confidence in WGS methods in clinical microbiology and molecular epidemiology.

WGS quality metrics encompass read length, library insert size, PHRED scores, and base pair composition. The PHRED score system assigns a quality, q (q =  − 10⋅ log10(p)), to each base in a WGS data file, denoting the probability of a miscall (Ewing & Green, 1998). At present, Illumina platforms dominate the market of short-read WGS technologies. Low PHRED scores are the most common quality issue encountered in Illumina data. Average quality scores typically decrease towards the 3′ends of reads (Dohm et al., 2008; Minoche, Dohm & Himmelbauer, 2011). In addition, Illumina reverse reads exhibit lower overall quality relative to forward reads (Guo et al., 2012; Tan et al., 2019). Given the correlation between PHRED scores and base miscalls, low PHRED scores are associated with more fragmented genome assemblies (Smeds & Künstner, 2011) and less reliable SNP discovery (Guo et al., 2012). Library insert size (also referred to as “insert length” in this manuscript) is another WGS-quality metric shown to be critical to bacterial genomics (Chen et al., 2015; Huptas et al., 2016). When insert lengths are shorter than read lengths, adapter sequences may occur in the WGS data set and average read length will be reduced (Turner, 2014). For bacterial genome assembly, increased insert lengths provide less fragmented, higher quality assemblies (Chen et al., 2015; Clooney et al., 2016; Head et al., 2014; Huptas et al., 2016; Trivedi et al., 2014). By contrast, insert lengths have shown less predictable impacts on SNPs analyses. One study employed short inserts with overlapping reads to enhance SNP calling accuracy (Chen-Harris et al., 2013). On the other hand, large insert sizes help identify genomic insertions, deletions, and rearrangements (Guan & Sung, 2016).

Read healing comprises any processing method applied to WGS reads to improve overall quality prior to their use in SNPs, genome assembly, or any other computational analysis. To mitigate the effects of low PHRED quality and other artifacts in raw WGS reads, dozens of open-source software tools have become available to perform read healing (Fabbro et al., 2013; Heydari et al., 2017; Patel & Jain, 2012). There are three broad categories of read-healing tools, filtering, trimming, and error correction (Smeds & Künstner, 2011). The filtering strategy removes entire reads failing a given quality threshold. The Illumina chastity filter is an example of the filtering approach (Whiteford et al., 2009). By comparison, trimming removes portions of reads which fail quality thresholds or distance thresholds from the 5′/3′ends of the reads. Prinseq and Fastx-toolkit are examples of tools implementing filtering, trimming, or both (Hannon, 2010; Patel & Jain, 2012; Schmieder & Edwards, 2011). The error correction strategy employs kmer or alignment techniques to correct miscalls in reads. Quake and BayesHammer provide examples of read healing by correction (Kelley, Schatz & Salzberg, 2010; Nikolenko, Korobeynikov & Alekseyev, 2013).

While many previous studies have compared read healing tools, few have characterized how quality metrics and read healing correlate within bacterial WGS sets representing outbreaks. The current study describes effects of quality metrics and read healing upon the Illumina WGS data representing four enteric bacterial outbreaks. This study also describes correlations between WGS quality, SNPs analysis, and genome assemblies for the outbreak data sets (Table S1). Admittedly, the results of this study are limited to neutral GC content organisms and do not include high or low GC content organisms such as Listeria. Filtering and trimming approaches to read healing elucidate the WGS quality metrics imparting the greatest effect upon intermediate analyses. Finally, the most effective healing techniques are recommended for addressing quality artifacts and improved intermediate analyses.

Materials & Methods

Isolates from four outbreak-associated Enterobacterales clusters were evaluated. All sequencing was performed on the Illumina MiSeq (San Diego, CA) using the Nextera XT (Illumina) library preparation. Isolates were collected over the span of several months to represent outbreak clusters and were submitted by multiple public health laboratories. As a consequence, 300-, 500- and/or 600-cycle sequencing chemistries were represented in each data cluster. The four outbreak clusters as represented Escherichia coli O26 (Cluster 1, Table S1A), Salmonella enterica serovar Reading (Cluster 2, Table S1B), S. enterica serovar Pomona (Cluster 3, Table S1C, (Gambino-Shirley et al., 2018; Walters et al., 2016)), and Shigella sonnei (Cluster 4, Table S1D). These predicted outbreak clusters were chosen due to the presence of quality artifacts such as high proportions of ambiguous nucleotides. The use of GC-neutral Enterobacterales genomes were expected to minimize GC-composition artifacts (Laehnemann, Borkhardt & McHardy, 2016) for single nucleotide polymorphisms (SNPs) and assemblies analysis.

Sequencing reads were analyzed for quality. Raw reads were checked for contamination using the standard Kraken1 database in Kraken v1.0.0; no conflicting taxonomies were detected (Wood & Salzberg, 2014). Raw reads were down-sampled to yield near uniform genome coverage within each of the four clusters prior to PHRED score analysis using CG-Pipeline (Ewing & Green, 1998; Kislyuk et al., 2010) (Table S2). Reads with ambiguous nucleotides, Ns, in down-sampled isolates were counted using a custom python wrapper for prinseq-lite.pl (Schmieder & Edwards, 2011) (Table S2). Full-coverage raw reads were processed through each of seven read-healing pipelines, followed by down-sampling, and estimation of quality metrics. To control for sequencing coverage of isolates, all raw and healed E. coli O26 (Cluster 1) reads were down-sampled to 50X coverage, S. enterica serovar Reading (Cluster 2) reads were down-sampled to 65X, S. enterica serovar Pomona (Cluster 3) reads were down-sampled to 58X, and S. sonnei (Cluster 4) reads were down-sampled to 60X using CG-pipeline scripts (Table S2). Sets could not be normalized to higher coverages without excluding isolates from clusters.

Sequence reads from isolates in all four clusters were each healed through seven pipelines or else were left raw for comparison. Four of the pipelines were based upon a single computational step, either read length filtering, quality trimming, blunt-end trimming, or error correction (Table S3). The prinseq-lite wrapper, windowQualPrinseqLite_R1andR2.py (https://github.com/darlenewagner/NGS_Multi_Heal), was used to implement the pipeline ‘noNmin100’, which filtered out reads based upon length and presence of ambiguous nucleotides (Table S3). The fastx_trimmer wrapper script, fastxQualAdaptTrimmer_R1andR2.py, implemented the pipeline, ‘fastxOnly-3pr’, and trimmed a fixed number of bases from the 3’ ends of reads using FASTX-Toolkit (Hannon, 2010) (Table S3). The prinseq-lite wrapper, windowQualPrinseqLite_R1andR2.pl, also implemented trimming based upon PHRED quality and read length/ambiguous nucleotide filtering (prinseq in Table S3). The remaining single step read-healing pipeline, ‘BayesHammer’ implemented SPAdes v. 3.6.1 at the setting ‘–only-error-correction’ to perform a Bayesian sub-clustering error correction (Nikolenko, Korobeynikov & Alekseyev, 2013). Briefly, BayesHammer’s error correction is based upon the number of changes required to match aligned subsequences. The three remaining pipelines were comprised of two-steps. The pipeline, ‘noNmin100-3pr’, combined the filtering parameters of noNmin100 with the 3′blunt-end trimming of fastxOnly-3pr (Table S3). The pipeline, ‘prinseq-3pr’, combined the parameters of ‘prinseq’ with a 3′blunt-end trim. A related pipeline, ‘prinseq-5pr3pr’, had the same parameters as prinseq-3pr but with an additional 5′trim from both R1 and R2 reads (Table S3). All of our trimming pipelines consist of removal of 3′and/or 5′ends of reads, which might be presumably lower-quality. In summary, six of the healing pipelines healed reads based upon three quality metrics: PHRED quality, ambiguous nucleotide composition, and read length. The seventh pipeline, BayesHammer, healed reads through correction of base miscalls (Nikolenko, Korobeynikov & Alekseyev, 2013).

For each combination of outbreak clusters and read healing pipelines, read healing effects on high-quality single nucleotide polymorphisms (SNPs) analysis were evaluated. Multiple sequence alignments based upon genomic SNPs were constructed using the Lyve-SET v1.1.4f (Katz et al., 2017) and the CFSAN SNP Pipeline version 1.0.1 (Davis et al., 2015). For the E. coli O26 (Cluster 1) isolates, the closed chromosome of strain 11368 (accession AP010953) was used as the Lyve-SET and CFSAN SNP Pipeline reference. For Salmonella Reading (Cluster 2), no closely related finished reference was available, so an assembly from one of the study isolates, CVM_N17S1020, was used as an internal reference. For the Pomona cluster, the PacBio chromosome, 2012K-0678 (accession CP020718), served as reference. For the Shigella cluster, the PacBio chromosome, 2015C-3794 (accession CP022455), served as reference. Prophage regions were detected using PHASTER (Arndt et al., 2016) and regions flagged as phage-derived were masked using the custom script, mask_prophages_within_chromosome.pl (https://github.com/darlenewagner/NGS_Multi_Heal). Masked prophage regions used in the three closed reference chromosomes in the study are shown in Table S4. For all four clusters, Lyve-SET filtering parameters were set to remove SNPs with less than 20X coverage, less than 95% consensus, and occurring within less than 5 bp of neighboring SNPs. In the CFSAN SNP Pipeline, the default filtering parameters were employed, 8X coverage and 90% consensus (Davis et al., 2015). Proportions of reads mapped to references were obtained using ‘flagstat’ in samtools−0.1.19 (Li et al., 2009). True positive SNPs count for method i, TPi, were estimated from the intersection of SNPs positions healed by three other methods, Sj, Sk, and S l, and the intersection of method i, S i.

TPi = — Si ∩ (S j ∩ Sk ∩ S l) —

False discoveries for method j, FPj, were estimated by taking the set difference between SNP positions of method j, Sj, and the union of SNP positions in all other methods.

FPj = — Sj –(S1 ∪…∪ Si≠j ∪ …∪ S7) —

Draft assemblies were built for each sequenced isolate using raw reads and the seven sets of healed reads. Initial assemblies were performed in SPAdes v3.6.1 (Bankevich et al., 2012) using four bash wrapper scripts each with ‘careful’ assembly and differing k-mer combinations. The wrapper scripts employed seven (-k 21, 35, 45, 55, 75, 95, 121), eight (-k 21, 33, 43, 51, 63, 73, 99, 123) or nine (-k 21, 33, 45, 55, 63, 77, 99, 105, 121 and 21, 35, 49, 57, 63, 77, 99, 105, 121) k-mer bin combinations. In this manner, raw and healed reads of each isolate was assembled four times; the assembly with the lowest number of contigs with lengths >500 bp was selected using the custom script, multiFastaContigAvgJudgement.py (https://github.com/darlenewagner/NGS_Multi_Heal). Assemblies of equal contig count were selected based upon higher average contig length; assemblies of equal average contig length were selected based upon higher N50. Raw or healed reads were mapped using SMALT v 0.7.6 (Ponstingl & Ning, 2010, https://www.sanger.ac.uk/science/tools/smalt-0) with settings as employed previously (Katz et al., 2017). Insert sizes were inferred from SMALT mappings of raw or healed reads onto their respective best-of-four draft SPAdes assemblies. Read lengths and pair distances were extracted from resulting SAM files using samtools (Li et al., 2009). An additional script, cLikeFastaStats.py (https://github.com/darlenewagner/NGS_Multi_Heal) was used to calculate assembly quality metrics such as contig count, N50, and maximum contig length. To compare the best SPAdes assembly against an alternate assembler, each isolate was assembled once each in Skesa using default parameters (Souvorov, Agarwala & Lipman, 2018). Improvements in read metrics, mapping metrics, and assembly metrics were evaluated using the Kruskal–Wallis rank sum test and Dunn post-hoc tests in the R packages, dplyr and FSA, respectively. To adjust p-values for multiple comparisons, the Benjamani-Hochberg correction was used through the FSA R package. Spearman correlation (rs) was computed using the R base package. All quality metrics were evaluated in comparison to raw sequence reads for the individual isolate.

Results

Quality metrics in raw and healed reads

Paired-end Illumina reads from isolates within Cluster 1 (E. coli) varied widely with respect to quality metrics (Table 1, Data S1). Eighteen of the isolates, sequenced using 500-cycle chemistry (Fig. 1), had read lengths ranging from 35 to 250 bp (average = 212.4 bp). Median insert sizes for 500-cycle reads ranged from 182 bp up to 371 bp (average of medians = 273.4 bp). Average read lengths tended to increase with increasing median insert lengths among isolates sequenced on 500-cycle chemistry with a robust Spearman correlation (rs = 0.792). Forward (R1) reads sequenced by 500-cycle chemistry had average PHRED qualities of 34.9 (σ = 0.73). Reverse reads (R2) exhibited lower PHRED scores, averaging 30.6 (σ = 2.1), with one isolate as low as 26.0 (Table 1, top left). PNUSAE001785 had at least one Ns position on 10.3% of R2 reads while 2014C-3572 had Ns positions in 3.4% of both R1 and R2 reads (Data S1). R1 and R2 PHRED scores both exhibited a weak correlation with ambiguous nucleotide composition (rs = −0.0596 and −0.180). The remaining two isolates in the O26 cluster, PNUSAE001782 and PNUSAE001783, were sequenced using 600-cycle chemistry (Fig. 1). The 600-cycle read lengths ranged from 35 bp to 300 bp (average = 224.5 bp) with R2 PHRED quality averages around 27.7. The 600-cycle isolates showed the lowest median insert lengths in the cluster, 191 bp for PNUSAE001782 and 161 bp for PNUSAE001783 (Table 1).

Table 1 Read quality metrics: R1/R2 PHRED quality, median insert lengths, and percentage of R1 + R2 reads with Ns.

Range of averages quality scores for reads from isolates and inserts; medians are shown in parentheses. Results of healing pipelines are shown per data set. noNmin100 not shown due to lack of statistically significant results when compared to raw reads.

		E. coliser. O26 (Cluster 1)	S. entericaser. Reading (Cluster 2)	S. entericaser. Pomona (Cluster 3)	Shigella sonnei (Cluster 4)	
raw reads	R1 qual.	33.0–36.1 (34.8)	32.5–35.7 (34.2)	32.2–36.9 (35.3)	34.2–36.8 (35.9)	
R2 qual.	26.0–33.5 (30.3)	27.5–32.8 (30.2)	26.1–35.3 (32.2)	30.2–35.4 (33.9)	
insert (bp)	161–371 (264)	175–531 (290)	184–556 (383)	212–492 (309)	
R1 + R2 Ns %	0.0–5.1	0.0–4.6	0.0–31.9	0.0 –12.2	
fastxOnly-3pr	R1 qual.	33.1–36.1 (34.8)	32.8–35.3 (34.4)	32.3–36.9 (35.3)	34.3–36.9 (35.9)	
R2 qual.	26.5–33.8 (30.8)	27.6–32.8 (30.4)	26.8–35.6 (32.6)	30.5–35.6 (34.2)	
insert (bp)	171–372 (268)	176–532 (293)	198–558 (389)	213–493 (310)	
R1 + R2 Ns %	0.0–4.7	0.0–4.5	0.0–31.9	0.0–3.2	
prinseq	R1 qual.	34.4–36.7 (35.7)	34.5–36.7 (35.4)	33.9–37.2 (36.0)	35.0–37.2 (36.5)	
R2 qual.	30.8–35.0 (32.9)	31.4–34.6 (33.2)	30.8–36.2 (34.2)	33.0–36.2 (35.3)	
insert (bp)	192–382 (283)	187–539 (296)	229–564 (406)	219–494 (312)	
R1 + R2 Ns %	0.0–3.5	0.0–3.2	0.0–1.5	0.0–3.7	
Prinseq-3pr	R1 qual.	34.5–36.7 (35.8)	34.5–36.7 (35.5)	34.1–37.2 (36.1)	35.0–37.2 (36.5)	
R2 qual.	31.1–35.2 (33.2)	31.6–34.8 (33.4)	31.3–36.3 (34.4)	33.1–36.3 (35.4)	
insert (bp)	192–383 (283)	187–546 (296)	229–564 (406)	220–494 (312)	
R1 + R2 Ns %	0.0–3.4	0.0–3.2	0.0–1.5	0.0–2.3	
prinseq-5pr3pr	R1 qual.	34.5–36.7 (35.8)	34.6–36.8 (35.6)	34.1–37.2 (36.1)	35.0–37.3 (36.5)	
R2 qual.	31.1–35.2 (33.2)	31.6–35.2 (33.6)	31.3–36.3 (34.4)	33.1–36.3 (35.5)	
insert (bp)	190–380 (281)	183–538 (293)	227–562 (404)	218–492 (310)	
R1 + R2 Ns %	0.0–3.4	0.0–3.1	0.0–1.4	0.0 –2.3	
bayesHammer	R1 qual.	33.1–36.1 (34.9)	32.6–35.7 (34.2)	32.3–36.9 (35.3)	34.2–36.9 (35.9)	
R2 qual.	26.8–33.6 (30.5)	27.6–32.8 (30.3)	26.3–35.4 (32.3)	30.4–35.5 (34.0)	
insert (bp)	189–412 (290)	182–540 (300)	227–587 (401)	216–499 (314)	
R1 + R2 Ns %	0.0–1.7	0.0–0.6	0.0–32.0	0.0–6.5	

Figure 1 Illumina Read Chemistries Used in the Study.

Forward reads (R1, in blue) and reverse reads (R2, red) with range of lengths and average lengths found in raw reads. Insert sizes were inferred from SMALT mapping to draft genome assemblies and are given here as per isolate median lengths between 5′ position of R1 and the 5′ position of R2.

After healing, R1 and R2 PHRED scores, along with ambiguous nucleotide count, showed significant improvement. Together, prinseq, prinseq-3pr, and prinseq-5pr3pr significantly improved R1 scores and R2 scores, indicated by Dunn test post-hoc p-values under 0.05 (Table S5, Data S1, df = 7). Overall, the prinseq, prinseq-3pr, and prinseq-5pr3pr pipelines increased average R2 quality by up to three points (Table 1). Ambiguous nucleotide content significantly decreased under prinseq, prinseq-5pr3pr, and prinseq-3pr (Table S5). The Dunn test indicated average read lengths significantly decreasing under prinseq, prinseq-3pr, and prinseq-3pr5pr (2.81 × 10−5 <  p < 6.43 × 10−4). For example, prinseq-5pr3pr decreased average read lengths by 25 bp relative to raw reads. No healing method significantly changed insert lengths. In summary, prinseq, prinseq-3pr, and prinseq-5pr3pr combined improved PHRED scores with a decrease in read lengths.

Paired-end reads from isolates of Cluster 2 (S. enterica serovar Reading) exhibited quality metrics values comparable to Cluster 1. R1 PHRED quality averaged around 34.2 while R2 PHRED quality averaged roughly 4 points lower (Table 1, Data S2). Out of twenty-one isolates in the Reading cluster, twelve had R2 PHRED quality scores below 30.0 (mean of averages = 30.2, σ = 1.5, Table 1). Median insert sizes per isolate varied from 175 bp to 531 bp (average = 279, σ = 86.1, Table 1). Thirteen isolates in the cluster had median insert lengths below 300 bp, including the four isolates sequenced by 600-cycle chemistry (218 bp to 253 bp). Average read lengths for 500-cycle chemistry ranged from 181 bp to 245 bp, while 600-cycle chemistry read lengths ranged from 228 bp to 245 bp. Isolates with longer read lengths also tended to have longer insert lengths, indicated by a moderate correlation (rs = 0.604). Counting only the 500-cycle chemistry reads, this correlation increased to r s = 0.953. Six of the Reading cluster isolates, including three of the isolates sequenced with 600-cycle chemistry, had Ns present on more than 1.0% of R1 and/or R2 reads (Data S2). R2 PHRED exhibited a slight positive correlation with percentage of Ns in R2 reads (rs = 0.301), while R1 PHRED had a negligible association with R1 Ns (r s = 0.0565).

Read healing in the Cluster 2 yielded improvement in R1/R2 PHRED and percent ambiguous nucleotides. Prinseq-3pr and prinseq-5pr3pr yielded the largest R2 PHRED score increases (mean of averages = 33.4 and 33.6, Table 1); for 12 out of the 21 isolates, these pipelines raised R2 scores by at least 4 points. Only prinseq, prinseq-3pr, and prinseq-5pr3pr significantly raised both R1 and R2 PHRED scores under the Dunn post hoc test (Table S5, df = 7). Insert lengths showed no improvement under the Kruskal–Wallis test. The Dunn post hoc test indicated that three out of the seven pipelines, prinseq, prinseq-3pr, and prinseq-5pr3r, significantly reduced read length (2.570 × 10−6 <  p < 2.601 × 10−5; df = 7). For instance, prinseq-5pr3pr reduced read length by an average of 33 bp below the 221 bp average for raw reads. In this manner, healing pipelines which increased R1 and R2 PHRED scores also reduced read lengths.

Paired-end reads for isolates of Cluster 3 (S. enterica serovar Pomona) exhibited the widest variation in quality metrics among all clusters analyzed. Average R2 PHRED quality across the 25 Pomona isolates ranged from 26.1 to 35.3 (mean = 32.2, σ = 3.1, Table 1, Data S3). R1 PHRED quality averaged 4 points higher (mean = 35.3, σ = 1.6, Table 1) than R2 scores. Median insert sizes varied widely from 182 bp to 556 bp (mean = 383.4, σ = 107.4, Table 1). Average read length and median insert length showed a moderate correlation among the twenty isolates sequenced with 500-cycle chemistry (rs = 0.693). The two isolates sequenced with 600-cycle chemistry, PNUSAS002454 and PNUSAS002455, showed robust read and insert lengths (avg. = 292 bp and 411 bp), yet exhibited the lowest R2 PHRED qualities (avg. = 26.3) for the cluster. The three isolates sequenced with 300-cycle chemistry (Fig. 1) had the highest R2 PHRED scores, ranging from at 35.3 to 36.6 (Table 1). At the same time, the 300-cycle sequenced isolate, 2015EL-1657B, had ambiguous nucleotides (Ns) on 64% of its R2 reads (Table 1, Data S3). Ns counts were high on two of the 500-cycle sequences: 2016K-0057 had 3% of R1 reads with Ns and PNUSAS002458 had 10% of R2 reads with Ns. R1 and R2 PHRED scores exhibited weak correlations with Ns content (rs = −0.304 and −0.114).

R2 PHRED quality and ambiguous nucleotide percentage were the only quality metrics in the Pomona cluster to show significant improvement. Only prinseq, prinseq-3pr, and prinseq-5pr3pr significantly improved R2 PHRED under Dunn’s post hoc test (0.0151 < p <  0.0230, Table S5). Prinseq-5pr3pr and prinseq-3pr both increased average R2 PHRED scores across the cluster by over 2 points, to 34.4 (Table 1). Ambiguous nucleotide percentages also significantly decreased after prinseq, prinseq-5pr3pr, and prinseq-3pr healing (1.66 × 10−6 < p < 2.30 × 10−6). Prinseq-3pr reduced PNUSAS002458 reads N-counts from 10.0% to 1.4% and reduced 2015EL-1657B (300-cycle) N-counts from 63% to less than 1%. Neither R1 PHRED nor median insert lengths exhibited significant improvements under the Kruskal–Wallis test (df = 7, Table S5). Prinseq-3pr, and prinseq-5pr3pr reduced average reads lengths to 197 and 198 bp, down from the 224 bp average for raw reads. This change was significant under the Dunn’s post hoc test (3.49 × 10−4 < p < 4.81 × 10−4). In this manner, improvements in read quality were accompanied by decreases in read lengths.

Paired-end reads of Cluster 4 (S. sonnei) exhibited robust R1 and R2 PHRED quality metrics but wide variability in read lengths, insert lengths, and ambiguous nucleotide content. Average R2 PHRED quality across the 21 Cluster 4 isolates ranged from 30.2 to 35.4 (mean = 33.9, σ = 1.4, Table 1, Data S4), while R1 PHRED scores averaged higher (mean = 35.9, σ = 0.3, Table 1). Median insert sizes ranged from 212 bp to 492 bp (mean = 309.3 bp; σ = 86.6 bp, Table 1) with 11 isolates below 300bp. All Cluster 4 isolates were sequenced by 500-cycle chemistry (Fig. 1) and had average lengths ranging from 191.5 bp to 245.7 bp. Average read lengths and median insert lengths were strongly correlated (r s = 0.964). Four isolates had ambiguous nucleotides (Ns) on more than 1% of reads. Ns occurred on 24% of R2 reads in PNUSAE013260 (Table 1) while Ns ranged from 2.6% to 3.3% on R1 reads in PNUSAE014640, PNUSAE014641, and PNUSAE014642 (Data S4). R1 and R2 PHRED scores showed only a weak, negative association with ambiguous nucleotides (rs = −0.337 and −0.260). On the other hand, longer reads and inserts tended to be associated with fewer overall ambiguous nucleotides (rs = −0.628 and −0.678). WGS reads of Cluster 4 thus showed close associations between read length, insert length, and nucleotide content quality metrics.

In Cluster 4, both R1 and R2 PHRED showed significant improvements under three of the seven read healing pipelines. Prinseq raised average R2 PHRED scores to 35.3 while prinseq-3pr and prinseq-5pr3pr both raised R2 scores to 35.6 (Table 1). Under Dunn’s post hoc test, prinseq, prinseq-3pr, and prinseq-5pr3pr yielded significant gains in R1 and R2 PHRED scores (Table S5, df = 7). Insert lengths were not significantly changed by any of the read healing pipelines (Table S5). All pipelines except noNmin100 and noNmin100-3pr significantly reduced read lengths (all p < 0.00941). For instance, prinseq-5pr3pr reduced average read length by 14 bp. As with the other three outbreak clusters, decreased read lengths accompanied increased PHRED scores.

Read healing and quality metrics impacting SNPs analysis

In Cluster 1 (E. coli), raw read quality metrics did not consistently predict read mapping as a quality metric; yet, raw reads enabled robust SNPs discovery. In Lyve-SET, mapped reads per isolate ranged from 53.2% to 76.6% (average = 66.1%) while CFSAN SNP Pipeline mapped reads ranged from 65.6% to 82.2% (average = 74.9%, Data S1). Lyve-SET properly-paired mapped reads were lower, from 26.6% to 63.6% (average = 45.0%) while the CFSAN SNP Pipeline did not calculate properly-paired reads. There was a strong correlation (rs = 0.878) between R2 PHRED scores and total percentage of reads mapped to the reference. R1 PHRED correlated weakly with total percentage of mapped reads (r s = 0.232). R1 and R2 PHRED showed weak to moderate correlations with properly paired read mapping (r s = 0.203 and 0.689). Read length and insert length showed negligible correlation with total mapped reads (r s <0.0827) while they were both moderately correlated with properly paired reads (rs = 0.331 and 0.505). Mean R1 and R2 Ns percentage showed a moderate negative association with read mapping and proper pairing (r s = −0.497 and −0.487). SNPs based upon raw reads yielded a multiple sequence alignment of 1,007 positions in Lyve-SET and 2,331 in CFSAN (Fig. 2A). Cluster 1 raw reads recovered 97% of true positive Lyve-SET SNPs (Fig. 3A, y-axis) and 99% of true positive CFSAN SNPs Data S5). There was a 0% false discovery rate for Lyve-SET (Fig. 3A, x-axis) and a 0.6% false discovery rate for CFSAN (Fig. S5A).

Figure 2 MSA Lengths from Unambiguous SNPs.

(A) SNPs for E. coli O26 (Cluster 1) with Lyve-SET (dark grey bars) and CFSAN_SNP_Pipeline (light gray bars). (B) SNPs for Salmonella enterica Reading (Cluster 2) with Lyve-SET and CFSAN_SNP_Pipeline. (C) SNPs for S. enterica Pomona (Cluster 3) with Lyve-SET and CFSAN_SNP_Pipeline. (D) SNPs for Shigella sonnei (Cluster 4) with Lyve-SET and CFSAN_SNP_Pipeline. For both Lyve-SET and CFSAN across all four clusters, SNPs were counted from final multiple sequence alignments including positions with ambiguous nucleotides.

Figure 3 Read Healing Effects on SNPs Identification.

ROC-like plots of unique Lyve-SET SNPs (estimated false discovery rate in Data S5) compared to detected concordant SNPs or True Positive Rate (estimated sensitivity in Data S5). (A) E. coli O26 (Cluster 1). (B) S. enterica serovar Reading (Cluster 2). (C) S. enterica serovar Pomona (Cluster 3). (D) S. sonnei (Cluster 4). Estimated false discovery and True Positive Rate for the CFSAN SNP Pipeline are plotted in Fig. S5.

Read-healing pipelines increased numbers of Cluster 1 SNPs detected by Lyve-SET while providing less consistent increases to CFSAN SNP. Percentage of Lyve-SET properly paired reads from all healing pipelines except noNmin100 exhibited significant improvements under the Dunn post hoc test (Fig. S1B). For instance, prinseq-3pr and prinseq-5pr3pr raised properly paired mapped reads to averages around 72.4–72.5%. Under prinseq-3pr and prinseq-5pr3pr total mapped reads ranged from 67.4% to 80.9% in Lyve-SET and 71.0% to 84.3% in the CFSAN SNP Pipeline (Data S1). In Lyve-SET, Prinseq-3pr produced a multiple sequence alignment (MSA) of 1,100 SNPs positions while Prinseq-5pr3pr yielded an MSA of 1,099 SNPs positions (Fig. 2A, (Data S5). Prinseq-3pr5pr identified 100% of concordant Lyve-SET SNPs (True Positive Rate in Fig. 3A) with only 0.2% of SNPs as predicted false discoveries (False Discovery Rate in Fig. 3A). Only prinseq produced a noticeable improvement in CFSAN SNP Pipeline results, with 2,402 SNPs (Fig. 2A) and a false discovery rate near 0.4% (Fig. S5A). Healed reads in both SNP pipelines produced 98% to 100% true positive SNPs, with 0.5% to 0.6% false discoveries (Fig. 3A; Fig. S5A; Data S5). False discoveries averaged 0.2% for Lyve-SET and were significantly greater at 0.6% for the CFSAN SNP Pipeline (pairwise Wilcoxon, p = 0.0078).

In Cluster 2 (S. enterica serovar Reading), raw read correlated with read mapping while the CFSAN SNP Pipeline outperformed Lyve-SET. Lyve-SET total mapped reads per isolate ranged from 54.4% to 92.8% (average = 78.5%, Fig. S2A), but properly paired mapped reads were as low as 34% per isolate (average = 54.5%, Fig. S2B). The CFSAN SNP Pipeline mapped reads ranged from 67.1% to 97.9% (average = 87.7%, Data S2). R2 PHRED quality scores showed robust associations with Lyve-SET total read mapping and properly paired mapping (r s = 0.758 and 0.669); R1 PHRED correlated less strongly with both (rs = 0.277 and 0.523). Ns content and Lyve-SET read mapping showed a weak correlation (r s = 0.332) while read length and insert length bore a negligible correlation with read mapping and properly paired mapping (rs <—−0.1844755—). In Lyve-SET, the raw reads yielded 71 SNPs positions (Fig. 2B) while the CFSAN SNP Pipeline detected 155 SNPs (Data S2). Lyve-SET found only 88% of estimated true positive SNPs from raw reads, yet was free from false discoveries (Fig. 3B). At the same time, the CFSAN SNP Pipeline yielded 99% of estimated true positives with a false discovery rate of 0.7% (Fig. S5B).

All seven healing pipelines improved Lyve-SET mapping and SNPs counts for Cluster 2. Read healing also contributed to robust results of the CFSAN SNP Pipeline. Reads healed through fastxOnly-3pr, noNmin100-3pr, prinseq, prinseq-3pr, and prinseq-5pr3pr exhibited significant increases in properly paired reads (Fig. S2B); Lyve-SET total mapped reads improved only slightly, averaging no more than 87.3% (Fig. S2A; Data S2). Average total read mapping in the CFSAN SNP Pipeline reached a maximum of 90.3% for reads healed through prinseq (Data S2). In Lyve-SET, Prinseq-5pr3pr and prinseq-3pr detected 93 to 94 SNPs positions (Fig. 2B); 100% of these were true positives, albeit with false discovery rates of 3% (Fig. 3B). In the CFSAN SNP Pipeline, prinseq-3pr, prinseq, and prinseq-5pr3pr, each detected 162, 163, and 166 SNPs, respectively. True positives of these CFSAN SNP Pipeline positions ranged from 95% to 96% with 0% false discoveries. False discoveries were statistically equivalent between the SNPs pipelines (pairwise Wilcoxon, p = 0.79), with Lyve-SET averaging 0.7% and the CFSAN SNP Pipeline averaging 0.6% (Fig. 3B; Fig. S5B).

Cluster 3 (S. enterica serovar Pomona) exhibited robust correlations between raw read and SNPs quality metrics while CFSAN outperformed Lyve-SET in SNP discovery. Lyve-SET total mapped reads per isolate ranged from 41% up to 96% (average = 84.7%, Fig. S3A) while properly paired reads were as low as 18% (average = 66.5%, Fig. S3B). CFSAN SNP Pipeline mapped reads ranged from 87.8% to 97.7% (average = 95.6%, Data S2). R2 PHRED quality correlated strongly to mapping percentages and properly paired reads (rs = 0.873 and 0.857). R1 PHRED quality was likewise strongly correlated with properly paired reads (rs = 0.801). Read lengths and ambiguous nucleotide content exhibited weak associations with mapping of proper read pairs (rs = −0.173 and rs = −0.298); Median insert lengths bore a moderate association with proper read pairing (rs = 0.526). Cluster 3 raw reads aligned in Lyve-SET to yield 412 SNPs (Fig. 2C) while the CFSAN SNP Pipeline detected 709 SNPs. Lyve-SET SNPs exhibited a true positive rate of 98%. The CFSAN SNP Pipeline true positives were slightly higher at 99% while both SNPs pipelines showed a 0% false discovery rate (Fig. 3C, Fig. S5C, Data S5).

Five out of the seven healing pipelines improved SNPs analysis quality across multiple metrics for Cluster 3. With respect to properly paired read mapping, only prinseq, prinseq-5pr3pr, and prinseq-3pr yielded significant improvements under the Dunn post hoc test (Table S6) where properly paired mapping rates ranged from 66% to 97% (average = 90.1%). Prinseq increased total read mapping to similar ranges for both Lyve-SET and the CFSAN SNP Pipeline, 86.8%-96.4% and 88.3%-98.0%, respectively (Data S3). Prinseq-3pr-healed reads yielded the largest raw number of SNPs positions in Lyve-SET, 443 in all (Fig. 2C), and enabled identification of 100% of the true positive positions with a 0% false discovery rate (Fig. 3C). All healing pipelines improved CFSAN SNP Pipeline SNPs discovery. Notably, prinseq-healed reads yielded 746 SNPs in the CFSAN SNP Pipeline (Fig. 2C) with a false discovery rate of 2.5% (Fig. 3C). False discoveries averaged 0.2% for Lyve-SET and 0.6% for the CFSAN SNP Pipeline, yet these rates were statistically-equivalent (pairwise Wilcoxon, p = 0.42).

Cluster 4 (S. sonnei) SNPs quality metrics correlated with only two of the read quality metrics, read length and insert length. In Lyve-SET, Cluster 4 isolate reads mapped to the reference from a 58.9% up to 82.5% (average = 68.9%, Fig. S4A). Properly paired raw reads mapped at much lower proportions, from 45.7% to 73.1% (average = 54.4%, Fig. S4B). CFSAN SNP Pipeline mapped reads ranged from 66.4% to 91.4% (average = 77.4%, Data S4). In contrast to Clusters 1-3, Cluster 4 reads exhibited little correlation between R1/R2 PHRED quality and percentage of mapped reads (rs = −0.210 and −0.169). Average read lengths, median insert lengths, and Ns content all had negligible correlations with read mapping percentage (−0.190 <rs <0.168). The Cluster 4 read lengths and insert lengths were strongly correlated with properly paired mapped reads (rs = 0.799 and 0.775). Raw reads of Cluster 4 aligning in Lyve-SET yielded 193 SNPs while the CFSAN SNP Pipeline more than tripled the number of SNPs to 608 (Fig. 2D). Both the Lyve-SET and CFSAN SNP Pipeline showed a true positivity rate of 99% and a 0% false discovery rate (Fig. 3D).

For Lyve-SET in Cluster 4, all read healing pipelines except noNmin100 and bayesHammer improved SNPs analysis results; the CFSAN SNP Pipeline improved SNPs for all healed reads except noNmin100-3pr and prinseq-3pr. FastxOnly-3pr, noNmin100-3pr, prinseq-5pr3pr, and prinseq-3pr improved Lyve-SET properly-paired mapping to averages of 66% to 71% (Table S6). The CFSAN SNP Pipeline performed best with prinseq read healing yielding 629 SNP positions, and total read mapping ranging from 67.0% to 92.0% (Data S4). Prinseq-3pr-healed reads produced 205 SNPs in Lyve-SET (Fig. 2D), a true positive rate of 99%, and a false discovery rate of 2% (Fig. 3D). Prinseq-healed reads yielded distinct but favorable results for both Lyve-SET and the CFSAN SNP Pipeline; Lyve-SET detected 199 SNPs, a true positive rate of 99% and a false discovery rate of 0.5% (Fig. 3D); The CFSAN SNP Pipeline, yielded 629 SNPs, a true positive rate of 99%, and a false discovery rate of 2.5% (Fig. S5D, Data S5). False discovery rates, averaging 0.7% for Lyve-SET and 0.9% for the CFSAN SNP Pipeline, were statistically-equivalent for both SNPs pipelines (pairwise Wilcoxon, p = 0.67).

Read healing and quality metrics relationships to assemblies

Raw reads from isolates in Cluster 1 (E.coli) yielded wide variations in assembly qualities while none of the read metrics correlated strongly with assembly quality. On average, raw reads assembled in SPAdes to 331 contigs, with N50 of 81,037 bp, and with maximum contig lengths around 218,070 bp (Table 2). Raw reads assembled in Skesa to an average of 341 contigs while N50 and maximum contigs averaged 75% to 80% shorter in Skesa (Table S7). Despite its high percentage of reads with ambiguous nucleotides (≈ 4%), 2014C-3572 had the best assemblies, 227 contigs in SPAdes, 221 contigs in Skesa, and near-equal maximum contigs of 296,819 in SPAdes and 296,311 in Skesa. Isolate PNUSAE001736 had the most fragmented assemblies: 435 contigs and a 48,358 bp N50 in SPAdes with 511 contigs and a 28,311 N50 in Skesa. R1 and R2 PHRED had a weak correlation with SPAdes and Skesa contig counts (−0.345 <  rs <0.106, (Data S1). Associations between average Ns per read versus contig count, N50, and maximum contig were weak (−0.183 <  rs <0.063). Correlations between read length, and assembly metrics were weak across SPAdes and Skesa (−0.330 <  rs <0.235).

Table 2 SPAdes Assembly quality metrics average values.

Boldface underlined values indicate one-sided Dunn post-hoc test improvement over raw read scores at α = 0.05 level of significance (with Benjamani-Hochberg correction). Reads healed through pipelines; noNmin100 and fastxOnly-3pr did not produce assemblies with statistically significant improvements over any metric and are not shown.

	Metric (Kruskal–Wallis p)	Raw Reads	noNmin100-3pr	prinseq	prinseq -5pr3pr	prinseq-3pr	bayesHammer	
E. coli O26
(Cluster 1)
Kruskal–Wallis
df = 7	Contigs (1.95 × 10−10)	330.5	318.6	295.9(0.006916)	295.1 (0.006258)	297.1 (0.009774)	287.2 (0.001382)	
N50 (0.01364)	81,036.8	84,793.3	92,749.2	92,574.8	93,140.8 (0.04853)	92,391.1	
Maximum contig (0.06474)	218,070.1	213,638.3	245,939.0	239,557.8	250,158.6	251,436.1	
S. enterica ser. Reading
(Cluster 2)
df = 7	Contigs (0.6353)	96.0	91.8	77.5	76.8	76.5	86.0	
N50 (0.9972)	243,104.2	265,306.8	270,566.3	277,891.2	270,003.6	253,005.8	
Maximum contig (0.9552)	552,949.7	625,007.5	653,148.7	658,147.9	602,089.9	581,313.7	
S. entericaser. Pomona
(Cluster 3)
df = 7	Contigs (0.506)	44.5	44.2	35.5	35.3	34.7	35.6	
N50 (0.9714)	412,213.0	431,232.0	412,871.0	431,778.0	465,935.0	431,969	
Maximum contig (0.5936)	911,229.0	838,277.0	1,000,451.0	1,019,369.0	1,023,669.0	1,039,598	
Shigella sonnei
(Cluster 4)
df = 7	Contigs (0.5547)	449.8	448.2	445.1	448.0	446.2	443.4	
N50 (0.8932)	23,931.1	24,054.5	24,117.1	23,931.1	23,929.4	24,009.8	
Maximum contig (0.999)	89,893.8	89,378.5	90,046.7	89,870.1	89,632.6	89,352.8	

Four of the seven healing pipelines enabled significant improvements in Cluster 1 assemblies. Read healing significantly improved SPAdes assemblies under the Kruskal–Wallis test; reads healed through prinseq, prinseq-3pr, prinseq-5pr3pr, and BayesHammer were significant under Dunn’s post hoc test (Table 2). Prinseq-3pr also significantly improved SPAdes N50 under Dunn’s post hoc test (Table 2). Read healing showed no significant improvement in Skesa assemblies (Kruskal–Wallis test, Table S7). For SPAdes assemblies, reads healed through BayesHammer assembled to the lowest (i.e., least fragmented) average contig count of 287.2 and largest maximum contig, 251,436 bp, while reads healed through prinseq-3pr had the largest average N50 value at 93,141 bp (Table 2). For Skesa assemblies, BayesHammer yielded the largest improvement for all three metrics: 303 contigs, 75,148 bp N50, and 202,003 bp maximum contig (Table S7). Maximum contig lengths did not significantly increase; yet, in both SPAdes and Skesa average maximum contigs from BayesHammer-healed reads showed a 15% increase over raw read assemblies (Table 2).

The raw reads generated from Cluster 2 (S. enterica serovar Reading) isolates produced assemblies of widely variable quality and were correlated with R1/R2 PHRED scores. Raw reads assembled in SPAdes to an average 96.0 contigs, a 243,104 bp N50, and 552,950 bp maximum contig lengths (Table 2). Assembly metrics for Skesa represented vast underperformance relative to SPAdes (pairwise Wilcoxon, p < 0.00024), showing roughly twice the average contig count as SPAdes (Table S7). Two isolates showed exceptional assemblies in SPAdes: CVM_N17S1018, a 600-cycle sequence (Fig. 1), had 36 contigs and an N50 of 455,001 bp, while PNUSAS032349, a 500-cycle sequence, exhibited the longest N50, at 637,471 bp, and the maximum contig, at 1,787,855 bp. The most robust Skesa assembly, PNUSAS032349, had 24 contigs, at 396,938 bp N50 and as the maximum contig at 1,234,155 bp (Data S2). By comparison, APHI_17-10777 (average R2 PHRED = 27.5) had the most fragmented assembly with 266 contigs, an N50 of 28,822 bp, and a maximum contig length of 97,074. Skesa also performed poorly for APHI_17-10777, with 1,019 contigs, an N50 of 6,851 bp, and a maximum contig length of 29,736. For both SPAdes and Skesa, contig counts slightly decreased with increasing R1 and R2 PHRED scores (−0.412 >rs >0.437). R1 and R2 PHRED scores exhibited a moderate, positive association with N50 and maximum contig length (0.430 <r s <0.526) for both assembly methods. Insert lengths and ambiguous nucleotide composition showed negligible correlation with contig count, N50, and maximum contig length (−0.190 <rs <0.192).

Reads from isolates in Cluster 2 did not yield significant improvements in assembly quality under any of the seven healing pipelines. Under the Kruskal–Wallis test, contig counts, N50, and maximum contigs of healed read assemblies did not significantly vary from raw reads for either SPAdes (Table 2) or Skesa (Table S7). Nonetheless, reads healed through prinseq-5pr3pr showed the largest overall improvements in SPAdes assemblies; average contig count reduced to 77 (by 20%), N50 increased by 14%, to 277,891 bp, and average maximum contig increased by 19%, to 658,148 bp (Table 2). For Skesa assemblies, BayesHammer yielded the most improvement: contig counts decreased by 34% (to 128), N50 increased by 30% (to 152,297 bp), and maximum contigs increased by 14% (to 366,842 bp, Table S7). Proportional improvements also occurred for prinseq-3pr, where SPAdes contig counts decreased by 20% and Skesa contig counts decreased by 28%.

Cluster 3 (S. enterica serovar Pomona) yielded high quality assemblies using raw reads and showed moderate associations with read quality metrics. On average, Cluster 3 raw reads assembled in SPAdes to 45 contigs, 412,213 bp N50, and 911,229 bp maximum contig lengths (Table 2). Skesa assemblies in Cluster 3 exhibited poorer assembly metrics than SPAdes (pairwise Wilcoxon, p < 0.0263); the average N50 and maximum contig length had only 52% to 58% the length of corresponding SPAdes metrics (Table S7). The 300-cycle sequence (Fig. 1), 2015EL-1657B, yielded the most robust SPAdes assembly with 25 contigs and an N50 of 757,183, despite the occurrence of Ns on 63% of its R1 reads. The best Skesa assembly was PNUSAS002461 with 21 contigs and an N50 of 455,461 bp (Data S3). PNUSAS002458, with an R1 PHRED score of 29.8 and 10% Ns on its R2 reads, yielded the poorest SPAdes assembly, 151 contigs and an N50 of 220,687 bp. Through Skesa, PNUSAS002458 yielded 91 contigs and an N50 of 96,273 bp. The correlations between ambiguous nucleotides and contig count, N50, and maximum contig length were negligible (−0.0531 <rs <0.174). R1 and R2 PHRED showed a moderate inverse association with contig counts for SPAdes and Skesa (−0.493 <r s <-0.301). SPAdes/Skesa N50 and maximum contig length increased moderately with R1 and R2 PHRED (0.351 <rs <0.591). Insert lengths also increased with maximum contig lengths (r s = 0.5354). Otherwise, insert lengths and read lengths showed little association with contig counts or N50 (−0.3984 <rs <0.2554).

Read-healing pipelines made only slight improvements to assembly quality metrics for Cluster 3. Read healing did not significantly change contig count, N50, or maximum contig length for either SPAdes (Table 2) or Skesa (Table S7) under the Kruskal–Wallis test. Despite the absence of statistically robust improvements, prinseq-5pr3pr, prinseq-3pr, and BayesHammer provided proportional gains to assembly quality metrics. Prinseq-5pr3pr reduced average SPAdes and Skesa contig counts to 35 and 42, respectively; these comprised 20% and 15% reductions from the average raw read contig counts (Table 2 and Table S7). Prinseq-3pr increased SPAdes N50 to an average of 465,935 bp (a 13% increase, Table 2) and Prinseq increased Skesa maximum contig length to 558,596 bp (a 16% increase, Table S7).

Cluster 4 (S. sonnei) assemblies were considerably more fragmented than those of Clusters 1 through 3. On average, Cluster 4 raw reads in SPAdes assembled to 449.8 contigs, a 23,931 bp N50, and 89,893 bp maximum contig length (Table 2). Skesa assembled Cluster 4 into fewer contigs, 392 on average, significantly less fragmented than SPAdes (pairwise Wilcoxon, p = 0.0000317). At the same time, Skesa produced an average 23,494 bp N50 and an 88,610 bp maximum contig length (Table S7). Isolate PNUSAE013260 yielded the most fragmented SPAdes assembly with 501 contigs and an N50 of 19,019 bp; the Skesa assembly was somewhat less fragmented with 468 contigs and an N50 of 18,211 bp. Isolate PNUSAE013900 exhibited the best Cluster 4 assembly, with SPAdes yielding 408 contigs and a 25,378 bp N50; Skesa yielded 359 contigs and a 25,647 bp N50. For both assemblers, R1 and R2 PHRED scores bore only weak to moderate correlations with contig count, N50, or maximum contig length (−0.4144 <rs <0.2166, Data S4). For SPAdes assemblies, maximum contig length showed an unexpected decreasing trend with read length and insert length (r s = −0.6061 and −0.5866). For Skesa, maximum contig length also showed an unexpected increase with Ns content (rs = 0.506). Read length and insert length showed only weak correlations with contig count and N50 (−0.384 <rs <0.2462).

None of the read healing pipelines significantly improved any of the assembly metrics for Cluster 4 reads. Under the Kruskal–Wallis test, neither contig count, N50, nor maximum contig length could be improved by read healing by either SPAdes (Table 2) or Skesa (Table S7). A few proportional improvements were observed. BayesHammer-healed reads assembled to an average of 443.4 contigs in SPAdes and 386 contigs in Skesa; both represented a less than 2% decrease from raw read contig counts (Table 2). For SPAdes, prinseq-healed reads assembled to an average N50 of 24,117 bp (less than a 1% improvement, Table 2). Skesa lengthened average maximum contigs from prinseq-healed reads by 1%; At the same time, metrics did not consistently improve across Skesa assemblies of healed reads (Table S7).

Discussion

This study has shown that WGS quality metrics, particularly PHRED scores, have varying impacts upon analyses of bacterial genomes with average G+C content. As of August 2020, Illumina data at NCBI SRA (https://www.ncbi.nlm.nih.gov/sra) were available for 198,928 E. coli, 343,457 S. enterica, and 13,374 S. sonnei isolates. Given this volume of WGS data representing a wide variation in quality control standards, methods of defining and improving WGS quality are needed.

R1 and R2 PHRED scores exerted the strongest effects on SNPs analyses while insert length and read length effects on SNPs could not be ruled out. R1 and R2 PHRED scores exhibited positive correlations with the mapping stage of SNPs analyses in all except Cluster 4. Two of the three PHRED-quality-trimming-based healing methods, prinseq and prinseq-3, yielded enhanced SNPs discovery results in the CFSAN SNP Pipeline and Lyve-SET, respectively. High PHRED scores have previously been shown to increase proportions of mapped reads, which in turn increased numbers of SNPs discovered (Yu et al., 2012; Yu & Sun, 2013). By contrast, ambiguous nucleotide composition (Ns) for raw reads showed only a slight association with read mapping in Cluster 1 and none in Clusters 2 through 4. Moreover, the ambiguity-free reads processed by noNmin100 underperformed in SNPs identification relative to all other healed sets. Hence, ambiguous nucleotide composition is not a critical quality metric for Illumina WGS data of enteric bacteria. Insert length showed a positive association with properly-paired mapped reads in Cluster 4, and to a lesser extent in Clusters 1 and 3. Greater proportions of reads mapping in proper pairs, in turn, may have enabled identification of more SNPs. Read length was positively correlated with insert lengths among 500-cycle reads across the four clusters. The strong association between read lengths and insert lengths in Cluster 4 make it unclear which two quality metrics impacted SNPs analysis the most. Despite previous studies (Chen-Harris et al., 2013; Guan & Sung, 2016; Laehnemann, Borkhardt & McHardy, 2016) and this current study, further effort may be needed to show how insert sizes affect SNPs and how insert length, read length, and PHRED scores affect number and quality of SNPs.

Choice of read-healing strategy also impacted SNPs analyses. Five of the read healing pipelines, fastxOnly-3pr, noNmin100-3pr, prinseq, prinseq-5pr3pr, and prinseq-3pr, all employed trimming of 3′and/or 5′ends of reads. By contrast, noNmin100 employed filtering only and BayesHammer employed correction only. NoNmin100 filtering (no trimming) performed well for Lyve-SET in Cluster 2, but underperformed for the other clusters with respect to SNPs counts and/or true positive rate. The simple, blunt-end trimming, fastxOnly-3pr, underperformed for the CFSAN SNP Pipeline. Prinseq, which trimmed based upon PHRED quality, proved to be effective for increasing SNPs counts for both Lyve-SET and the CFSAN SNP Pipeline. Trimming was been shown to increase concordant (estimated true positive) SNPs in a study for human genome clinical Illumina data (Yu & Sun, 2013). At the same time, another study showed trimming to decrease numbers of true positive SNPs, albeit by only 0.7% (Liu et al., 2012). In our study, prinseq-healed reads enabled detection of a greater or equal amount of true positive SNPs compared to raw reads across all clusters except for Cluster 2 in the CFSAN SNP Pipeline. False discovery rates in our study reached a maximum of 3% for reads healed through prinseq-5pr3pr and prinseq-3pr (Cluster 2 only, Fig. 3B). These two healing methods produced less than 1.5% false discoveries in the other three clusters for both SNP pipelines. Indeed, prinseq-5pr3pr and prinseq-3pr yielded less than 1% false discoveries for the CFSAN SNP Pipeline. Such increases in SNPs analysis performance matches the significant increase in PHRED scores provided by prinseq-5pr3pr and prinseq-3pr.

The CFSAN SNP Pipeline detected larger numbers of SNPs relative to Lyve-SET, possibly due to the former’s less stringent coverage threshold of 8X (versus 20X in Lyve-SET). The CFSAN SNP Pipeline also mapped higher proportions of reads, as it employs Bowtie2 (Langmead & Salzberg, 2012), compared with the more stringent SMALT in Lyve-SET (Katz et al., 2017; Ponstingl & Ning, 2010). Given that Lyve-SET is designed to be more conservative, it is surprising that Lyve-SET and the CFSAN SNP Pipeline produced similar proportions of false discoveries for three out of the four clusters. In any case, the quality-based trimming methods, prinseq, prinseq-5pr3pr, and prinseq-3pr, plus the correction-based BayesHammer, enhanced SNPs analysis across both SNPs pipelines. Accuracy of SNPs detection may depend less upon choice of SNPs pipeline and more upon choice of read-healing and the peculiarities of the microbial cluster under consideration.

Contrasting with SNPs analysis quality, associations between read quality metrics and assembly quality were less consistent. R1 and R2 PHRED scores did show some association with reduced contig counts in the Salmonella clusters (2 and 3). The three quality-based trimming methods yielded significant improvements in Cluster 1 only. As was the case for SNPs, ambiguous nucleotide count (Ns) seemed to have no relationship to assembly quality. Unexpectedly, insert lengths were not strongly associated with robust assemblies; greater insert lengths correlated with maximum contig lengths in Cluster 3 only. A previous study suggested that higher insert average length and more variable insert lengths lead to less fragmented assemblies (Huptas et al., 2016). Yet, the variability in read length chemistry in the current study may have obscured any relationship between insert length, read length and assembly metrics. While PHRED scores positively impacted assembly quality, additional read quality metrics, not measured by this study, may also be relevant. Previous studies have shown genomic GC content bias (Chen et al., 2013; Huptas et al., 2016; Ross et al., 2013) and Illumina-specific substitution errors (Laehnemann, Borkhardt & McHardy, 2016) to impact genome assemblies. Cluster 4 assemblies exhibited seemingly intractable levels of fragmentation, never more contiguous than 354 contigs (i.e., Skesa after BayesHammer-healing). Cluster 4 represents S. sonnei, which exhibit highly-fragmented assemblies. The current study represents a modest improvement over Shigella assembly averages around 500 contigs in a previous study (Timme et al., 2018). The high incidence of insertion sequences, repeats, and recent rearrangements in Shigella spp. genomes likely make assembly of these organisms more challenging than other enteric bacteria (Page et al., 2016; Yang et al., 2005). Further investigation into optimizing read quality metrics for improved draft genome assemblies of Shigella spp. and other enterics may be needed.

The findings of this study are subject to several limitations given that it used real WGS data rather than simulated data. First, all the Illumina reads in this study were obtained via Nextera XT library kits (Quail et al., 2012; Syed, Grunenwald & Caruccio, 2009). Nextera XT libraries have well-documented quality issues, particularly G+C% bias (Sato et al., 2019; Tyler et al., 2016). The KAPA and TruSeq have been shown to produce higher quality bacterial genomic reads with less fragmentation of genome assemblies (Jones et al., 2015; Seth-Smith et al., 2019). Compared to Nextera XT, Nextera DNA Prep reads also assembled to higher quality draft genomes as measured by N50, contig count, and plasmid detection (Haendiges, Jinneman & Gonzalez-Escalona, 2021). For this study, reads from such higher-quality libraries would likely have behaved differently under the healing pipelines investigated. Second, Clusters 1 through 3 exhibited mixed Illumina sequencing chemistries, with small numbers of 300-cycle and 600-cycle analyzed along with the majority 500-cycle. The 151-bp 300-cycle reads in these clusters tended to have higher PHRED quality than 500-cycle or 600-cycle. By contrast, the 301-bp 600-cycle reads trended towards the lowest PHRED quality scores. While these trends in read cycle chemistry are not generalizable to other studies, they made analyses in this study more complex. Third, the choice of outbreak clusters in this study was restricted to enteric bacteria with neutral GC content (50% to 55%). Results for read and analysis quality metrics from this study may not be relevant to low-GC foodborne pathogens such as Listeria monocytogenes and Campylobacter spp. Fourth, the scope of quality metrics analyzed in this study did not include substitution nor indel errors for the Illumina reads. Previous investigations into Illumina sequencing quality found that PHRED scores are not always a reliable predictor of substitution/indel errors (Schirmer et al., 2016; Zhang et al., 2017). Future studies of enteric bacterial WGS quality may need to incorporate quality score recalibration into SNPs analysis to compensate for WGS errors and biases (DePristo et al., 2011; O’Rawe et al., 2013). Further SNPs validation of enteric bacteria would benefit greatly by incorporating a cluster of microbial genomes with validated or curated SNPs. Finally, assemblies in this study were measured only for contiguity, not correctness. A simulated read set derived from a finished genome could have provided metrics for assembly correctness, as employed in the Assemblathon (Earl et al., 2011). On the other hand, simulated data may not represent the errors or variation present in real WGS reads (Salzberg et al., 2012). While SPAdes has been demonstrated to be among the most accurate assemblers for small genomes (Huptas et al., 2016), Skesa showed favorable results in assemblies of S. sonnei (Cluster 4). Hence, limitations to this study are related to the chosen WGS data sets in this study: enteric bacteria linked to outbreaks.

Conclusions

WGS has greatly enriched evolutionary and molecular studies of enteric bacteria while expediting outbreak responses to these common and widespread pathogens. As WGS grows in importance to clinical bacteriology, additional computational methods, such as MLST, for outbreak prediction will develop further (Besser et al., 2018; Besser et al., 2019; Moran-Gilad et al., 2015; Page et al., 2017). Moreover, metagenomics approaches to WGS may become indispensable as culture-independent diagnostics of enteric infections becomes widespread (Besser et al., 2018). Future studies should elucidate WGS quality metrics effects upon MLST and metagenomics. In this study, multiple read-healing methods performed well for SNPs analysis; choice of read-healing method was shown to be more important than choice of SNP pipeline. However, the read-healing methods detailed in this study had less impact upon genome assembly. Thus, read-healing methods which improve SNPs analyses may or may not be effective for MLST or metagenomics. In future studies, MLST, metagenomics, or other WGS analyses may require customized preprocessing tools to yield optimal results. As an additional future study, misassemblies of genomes could be evaluated with respect to read quality metrics. Regardless of the specifics of WGS development, quality metrics such as PHRED score, read length, insert length, and various nucleotide compositional measurements will allow comparison between datasets and methodologies. As WGS capabilities expand, standards for these quality metrics must be evaluated and adapted for all stakeholders.

Supplemental Information

Supplemental Information 1 NCBI SRA identifiers, divided by cluster

Click here for additional data file.

Supplemental Information 2 Scripts used to estimate quality metrics and downsample WGS data files

Click here for additional data file.

Supplemental Information 3 Scripts used to implement read healing pipelines

All pipelines tested on isolates from Clusters 1 through 4. Ambiguous nucleotide content was evaluated using the script, countReadsWithAmbig.py, for all raw and healed reads. Custom scripts available for download from https://github.com/darlenewagner/NGS_Multi_Heal

Click here for additional data file.

Supplemental Information 4 Chromosomal coordinates masked for prophages prior to LyveSET mapping

No coordinate masking employed for S. enterica ser. Reading draft assembly, CVM_N17S1020.

Click here for additional data file.

Supplemental Information 5 Quality metrics and variation across read trimming/healing methods

R1/R2 PHRED quality, median insert lengths, and percentage of Ns in R1 + R2 reads across trimming/healing methods are indicated by range of values. Dunn test post-hoc p-values are given in parenthesis for each healing method’s comparison with raw reads. Significant p-values ( α < 0.05) in boldface.

Click here for additional data file.

Supplemental Information 6 Variation in lyveSET read-mapping Quality metrics

Total proportions of reads mapped to reference genomes and proportions of reads mapping with proper pairing. Dunn test post-hoc p-values are given in parenthesis for each healing method’s comparison with raw reads. Significant p-values ( α < 0.05) are in boldface.

Click here for additional data file.

Supplemental Information 7 Assembly quality metrics average values for Skesa

Click here for additional data file.

Supplemental Information 8 Escherichia coli O26 (cluster 1) with quality metrics

Click here for additional data file.

Supplemental Information 9 Salmonella enterica ser. Reading (cluster 2) with quality metrics

Click here for additional data file.

Supplemental Information 10 Salmonella enterica ser. Pomona (cluster 3) with quality metrics

Click here for additional data file.

Supplemental Information 11 Shigella sonnei (cluster 4) with quality metrics

Click here for additional data file.

Supplemental Information 12 Detailed SNP information

Click here for additional data file.

Supplemental Information 13 E. coli O26 (Cluster 1) Read Healing and Read Mapping

Total proportions of reads mapped to reference genome, AP010953 (Data S1, column M). Kruskal–Wallis p = 6.203 × 10−12 ( df = 7); prinseq, prinseq-5pr3pr, prinseq-3pr, and bayesHammer read mappings differ from raw reads by p < 0.05 under the pairwise comparisons post hoc test (Table S5). (B) Proportions of reads mapping with proper pairing (Data S1, column N) against reference genome AP010953. Kruskal–Wallis p = 2.20 × 10−16 ( df = 7); All healed reads except noNmin100 show improved proper paired mapping by p < 0.05 under the pairwise comparison post hoc test.

Click here for additional data file.

Supplemental Information 14 S. enterica ser. enterica Reading (Cluster 2) Read Healing and Read Mapping

Total proportions of reads mapped (Data S2, column M) to assembly of strain CVM_N17S1020. Kruskal–Wallis p = 0.000302 ( df = 7) indicates that none of the healing pipelines significantly improve total read mapping. (B) Proportions of reads mapping with proper pairing (Data S2, column N) against assembly of strain CVM_N17S1020. Kruskal–Wallis p = 7.956 × 10−12 ( df = 7) indicates all healed reads except noNmin100 and bayesHammer show an improvement proper paired mapping over raw reads by p < 0.05 under the pairwise comparison post hoc test.

Click here for additional data file.

Supplemental Information 15 S. enterica ser. enterica Pomona (Cluster 3) Read Healing and Read Mapping

(A) Total proportions of reads mapped (Data S3, column M) to the genome of strain 2012K-0678. Kruskal–Wallis p = 2.20 × 1016 ( df = 7) indicates only prinseq, prinseq-5pr3pr, and prinseq-3pr read mappings differ from raw reads by p < 0.05 under the pairwise comparisons post hoc test. (B) Proportions of reads mapping with proper pairing (Data S3, column N) against the genome of strain 2012K-0678. Kruskal–Wallis p = 2.20 × 10−16 (df = 7) indicates only prinseq, prinseq-5pr3pr, and prinseq-3pr proper pairing rates differ from raw reads by p < 0.05 under the pairwise comparisons post hoc test.

Click here for additional data file.

Supplemental Information 16 Shigella sonnei (Cluster 4) Read Healing and Read Mapping

(A) Total proportions of reads mapped (Data S4, column M) to the genome of strain 2015C-3794. Kruskal–Wallis p = 0.3180 ( df = 7) indicates that none of the healing pipelines significantly improve total read mapping. (B) Proportions of reads mapping with proper pairing (Data S4, column N) against the genome, 2015C-3794. Kruskal–Wallis p = 2.7530 × 10−13 ( df = 7) indicates that fastxOnly-3pr, noNmin100-3pr, prinseq-5pr3pr, and prinseq-3pr have proper pairing rates above raw reads by p < 0.05 under the pairwise comparisons post hoc test.

Click here for additional data file.

Supplemental Information 17 Read Healing Effects on SNPs Identification

ROC-like plots of unique CFSAN SNPs (estimated false discovery rate in Data S5) compared to detected concordant SNPsor True Positive Rate(estimated sensitivity in Data S5). (A) E. coli O26 (Cluster 1). (B) S. enterica Reading (Cluster 2). (C) S. enterica Pomona(Cluster 3). (D) Shigella sonnei (Cluster 4).

Click here for additional data file.

We thank Soledad Ulloa Urrutia and Fernández Ordenes of Instituto de Salud Pública de Chile, Tonya Mackie and Tod Stuber of the Animal and Plant Health Inspection Service, USDA, and the Enteric Diseases Laboratory Branch, CDC, for contributing WGS data for the study.

Additional Information and Declarations

Competing Interests

Author Contributions

DNA Deposition

Data Availability

Darlene D. Wagner is employed by Eagle Medical Services, LLC.

Darlene D. Wagner conceived and designed the experiments, performed the experiments, analyzed the data, prepared figures and/or tables, authored or reviewed drafts of the paper, and approved the final draft.

Heather A. Carleton and Eija Trees conceived and designed the experiments, authored or reviewed drafts of the paper, and approved the final draft.

Lee S. Katz conceived and designed the experiments, prepared figures and/or tables, authored or reviewed drafts of the paper, and approved the final draft.

The following information was supplied regarding the deposition of DNA sequences:

The sequences are publicly available at NCBI (Table S1).

The following information was supplied regarding data availability:

GitHub: https://github.com/darlenewagner/NGS_Multi_Heal.

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
