# Peer review of "Evaluating whole-genome sequencing quality metrics for enteric pathogen outbreaks"

_PeerJ, doi:10.7717/peerj.12446_

## Round 0.1 · original submission · Minor Revisions

I agree with both reviewers that this is a well-written and valuable contribution, and will require primarily minor revisions. Please address all of the included reviewer comments. The suggestion to include additional species relevant to food security is worth considering, although not necessary as you do note clearly that your analysis is restricted to enterics. If you decide to add these species, please revise your title and text appropriately. Also, I second the comments by Reviewer 2 to provide additional justification for using lyveSET over CFSAN or cgMLST.

Reviewer 1 ·

Basic reporting

The article overall is well written, offers a good list of references that are used in the context of the results observed by the authors. Figures, tables and data are well made.

Experimental design

The methods are well described and in detail. The research question, although overall significant, looses a bit of impact due to the choices of dataset and tools selected by the authors. Specifically, the authors selected four datasets, representing (i) one E. coli outbreak, (ii) two Salmonella outbreaks and (iii) one Shigella outbreak. These three species are some what related (in comparison to other species also relevant to food safety such as Listeria monocytogenes and Campylobacter) and share many characteristics not shared by these other foodborne pathogens, such as a large genome (>= 5 MB) and ~50% GC content (in comparison to smaller genome sizes and low %GC of L. monocytogenes and Campylobacter). Therefore, it seems inappropriate to generalize the results obtained in this study to these other important foodborne pathogens. This problem could have been easily "fixed" by inclusion of other species representing a more diverse genomic context. In addition, the authors selected lyveSET as a tool for assessing the impact of raw read quality in SNP discovery. Although this tool has shown its usefulness in SNP discovery applied to foodborne pathogens, it does not seem to be currently used by any public health or regulatory agency for real-life investigations using WGS data. It could have been more useful, for example, to know how raw read quality impact the SNP discovery using the CFSAN SNP pipeline, which was developed by FDA-CFSAN, and is currently used by this agency in outbreak/traceback investigations. Also, public health agencies in the US, currently rely on cgMLST for outbreak investigation and, therefore, learning the impact of raw read quality on cgMLST results would have been more useful.

A second, unrelated concern: did the authors check the raw sequence data (and/or assemblies) for contamination? Can you please describe?

Validity of the findings

The conclusions are well stated. However, I missed a comment on why and how the raw read quality impacts on assemblies and SNP discoveries. In the case of SNP discovery, is it because of read mapping not being successful or because of not passing the selected cutoffs (minimum number of reads mapping, % concordance, etc), or both? For assemblies, do reads with low quality generate misassemblies or just prevent assembly of larger contigs? Providing the readers with some explanation on this would be very helpful.

Additional comments

Line 92: fix last reference (Hannon 2010)
Lines 109-110: The causal-effect relationship presented in this sentence does not make sense to me. Why isolates collected over the span of several months necessarily would need to have their genomes sequenced using 3 different Illumina cycles?
Line 114: "...such as high..."
Lines 170 (in comparison to line 167): Why using a different notation here? I suggest using:
FPi = | Si - (Sj U Sk U Sl)|

Reviewer 2 ·

Basic reporting

The language is clear throughout. However, should the sentence in line 99 begin with "This study" instead of "The study", as the previous sentence refers to "the current study"?

The authors provide adequate explanations for the filtering and trimming read-healing strategies (lines 87 to 93), but the explanation of how error correction strategies work is lacking. How do kmer and alignment techniques correct miscalls?

The figure caption for Figure 2 appears to be incorrect, it appears to be a repeat of the caption for Figure 1.

Figures S1 and S3 are not referred to in the text.

Line 368 to 369 - Figure 3 is referenced, but this text is referring to Figure S3.

There were some errors in references cited in the text (e.g. line 92, is there supposed to be a semi-colon after the Schmieder & Edwards 2011 reference?; line 531, the dates are missing).

Experimental design

Methods are described in detail, including locations of scripts and flags used during analyses. The Benjamani-Hochberg correction is mentioned in the caption for Table 2, but not described in the methods section.

Validity of the findings

The authors describe in detail their findings and conclusions based on the data.

Although the authors explain that SPAdes is demonstrated to be among the most accurate assemblers, it would be interesting to see the impact of an assembler such as SKESA (Souvorov et al., 2018) which is another popular assembler that is more scrupulous than SPAdes and has been shown to produce higher quality assemblies.

Additional comments

Overall well written. As a note, line 53 states that genome assembly "enables" characterisation of genes involved in resistance and virulence, however assembly is not required for gene detection (e.g. SRST2 by Inouye et al., 2014; and the KMA tool by Clausen et al., 2018).

The article may also benefit from a description of how widespread SNP analyses are, and what they are commonly used for.

---

## Round 0.2 · accepted · Accept

The reviewer is happy with the minor corrections on the final manuscript. thanks for submitting to PeerJ

Reviewer 1 ·

Basic reporting

No comment. I thank the authors for clarifying and responding to my previous comments.

Experimental design

No comment.

Validity of the findings

No comment.

Additional comments

No comment.